# Optimizing CT Esophagography: Ex Vivo Study on Contrast Ratios, Image Quality, and Dual-Energy Benefits

**DOI:** 10.3390/bioengineering11121300

**Published:** 2024-12-20

**Authors:** Luwen Hao, Xin Chen, Yuchen Jiang, Yufan Wang, Xuemei Hu, Daoyu Hu, Zhen Li, Yaqi Shen

**Affiliations:** 1Department of Radiology, Tongji Hospital, Tongji Medical College, Huazhong University of Science and Technology, Wuhan 430030, China; haoluwen2020@163.com (L.H.); jiangyuchen@hust.edu.cn (Y.J.); wyf2712803@163.com (Y.W.); xmhu@hust.edu.cn (X.H.); daoyuhu@hust.edu.cn (D.H.); zhenli@hust.edu.cn (Z.L.); 2Department of Radiology, Taikang Tongji (Wuhan) Hospital, Wuhan 430050, China; sanjin45@163.com; 3School of Public Health, Medical College, Wuhan University of Science and Technology, Wuhan 430065, China

**Keywords:** CT esophagography, ex vivo experiments, oral contrast agent, image quality, dual-energy CT

## Abstract

This study aimed to optimize CT esophagography by identifying effective oral contrast dilution ratios and exploring the advantages of dual-energy CT (DECT) over conventional CT for improving image quality. Ex vivo experiments using iodine contrast agents (320–400 mgI/mL) at 21 dilution ratios were scanned at three voltages, with additional dual-energy scans generating various reconstruction images. Image quality was assessed both objectively and subjectively. The study found significant variability in image quality across different dilution ratios. Specific dilution ratios that produced image quality comparable to the control group (a commercial oral contrast agent) and those meeting the standards for clinical diagnosis and high-quality images were identified based on image quality assessments. Recommendations for preparing 100 mL of oral contrast solution were provided, such as for achieving high-quality images at a scanning voltage of 100 kVp: the optimal dilution ratios are 1:6 to 1:19 for 320 mgI/mL, and 1:8 to 1:19 for 350 to 400 mgI/mL. Additionally, beam-hardening artifacts were significantly reduced in DECT images. These findings provide valuable guidance for improving CT esophagography protocols.

## 1. Introduction

Esophageal perforation is linked to severe complications and carries a high risk of mortality, ranging from 10% to 90% [1]. The condition can result from various causes, including spontaneous rupture, the ingestion of foreign bodies, or iatrogenic injury. The mortality rate is significantly higher when treatment is delayed beyond 48 h compared to within the first 24 h [2]. Early diagnosis remains challenging, as evidenced by the fact that only about half of the cases are admitted to the hospital within 24 h of symptom onset [3]. The signs and symptoms of esophageal perforation are often nonspecific [4]. Early radiological diagnosis and accurate post-treatment assessment are crucial for improving patient outcomes, reducing hospital stay duration and costs, and enhancing quality of life.

The American College of Radiology (ACR, 2019) recommends fluoroscopic esophagography as the initial imaging method for detecting esophageal perforation [5]. However, the application of fluoroscopic esophagography in an emergency is often limited [6]. CT esophagography is increasingly utilized for the precise evaluation of esophageal perforation, and the use of an oral contrast agent improved the positive predictive value of CT examination [2]. CT esophagography can serve as a routine follow-up tool to evaluate suspected esophageal perforation and can obviate fluoroscopic esophagography [2,7]. The World Society of Emergency Surgery (WSES 2019) recommends CT esophagography for esophageal perforation [8]. However, a consensus on standardized CT esophagography protocols is lacking.

Currently, there is significant variability among institutions regarding the concentration specifications of iodine contrast medium and the dilution ratios employed for oral contrast preparation in CT esophagography [2,6,7,9,10,11,12,13,14,15,16,17,18,19,20,21]. Tube voltages of 80 kVp, 100 kVp, and 120 kVp are commonly used in chest CT scanning to accommodate patients with different body sizes [22]. Contrast medium concentration significantly influences image quality. High-concentration oral contrast increases beam-hardening artifacts [23], while low-concentration oral contrast compromises density contrast, affecting the visualization of perforations and adjacent structures. When iodine contrast medium with various concentration specifications is used for oral contrast and scanned at different tube voltages, it needs to be diluted with different proportions of saline.

Dual-energy CT scans, equipped with various tools including metal artifact reduction (MAR) [24], monoenergetic reconstruction [25], and virtual non-enhanced (VNE) reconstruction [26] can effectively reduce background noise and beam-hardening artifacts (*BHA*), thus improving overall image quality. Additionally, dual-energy CT provides more accurate material characterization compared to conventional CT [27], allowing for enhanced differentiation among substances such as iodine, calcium, and soft tissue. This advantage can be particularly beneficial for diagnostic imaging. However, the use of dual-energy CT to improve image quality for CT esophagography has not yet been explored. The aim of this study was to optimize the dilution ratio for the preparation of oral contrast by evaluating the image quality in a phantom study mimicking CT esophagography examination, to provide recommendations for the CT esophagography protocol, and try to evaluate the additional value of dual-energy CT in improvement of image quality.

## 2. Materials and Methods

### 2.1. Ex Vivo Study

Oral contrast agents were prepared by diluting four iodine-based contrast media (320 mgI/mL, 350 mgI/mL, 370 mgI/mL, and 400 mgI/mL) with normal saline in 21 different volume ratios (1:0, 1:1, 1:2, 1:3, 1:4, 1:5, 1:6, 1:7, 1:8, 1:9, 1:10, 1:11, 1:12, 1:14, 1:15, 1:17, 1:19, 1:24, 1:49, 1:99, and 1:199). These ratios were selected based on prior studies and clinical practices, which have reported a wide variability in dilution concentrations across different institutions [2,6,7,9,10,11,12,13,14,15,16,17,18,19,20,21] (Appendix A).

Two-layered tube-like phantoms (Figure 1) (iodine-contained tubes placed in hollow tubes in the core of a water tank) were constructed and underwent conventional helical CT scan and dual-energy CT scan. Detailed information about the experiment can be found in Appendix A.

### 2.2. Image Quality Evaluation

Objective assessment of image quality: Objective evaluations were conducted on the acquired images by measuring parameters related to CT attenuation and image noise. Three axial slices without air bubbles were selected from each Eppendorf tube by an observer. Four distinct regions of interest (ROIs) were segmented (Figure 1) independently by two radiologists using ImageJ software version 1.52a (National Institutes of Health, Bethesda, MD, USA),

ROIa: The contrast agent within the Eppendorf tube, representing esophageal contrast material with high CT attenuation.ROIb: The wall of the hollow plastic tube, mimicking other slightly high-density tissues in the mediastinum.ROIc: The transition zone between the contrast agent and the surrounding plastic tube, simulating the region adjacent to the esophagus in a clinical setting.ROId: The background region outside the tube, representing a reference background area required for image quality analysis.

The CT attenuation (Hounsfield Unit, *HU*) and *SD* (image noise, *HU*) of ROIs were measured. Objective parameters of image quality included Contrast-to-Noise Ratio (*CNR*), Signal-to-Noise Ratio (*SNR*), and Beam-Hardening Artifact (*BHA*). These parameters, which are commonly used for objective image quality assessment, were calculated using the following formulas [25,28]:

*CNR* was calculated to evaluate the contrast between specific regions of interest: *CNR_a/b_*: represents the contrast between esophageal contrast material (ROIa) and slightly high-density tissues (ROIb).
$${CNR}_{a/b}=\frac{{{HU}_{a}-HU}_{b}}{\sqrt{{{SD}_{a}}^{2}+{{SD}_{b}}^{2}}}$$

*CNR_a/c_*: represents the contrast between esophageal contrast material (ROIa) and adjacent tissues (ROIc).
$${CNR}_{a/c}=\frac{{{HU}_{a}-HU}_{c}}{\sqrt{{{SD}_{a}}^{2}+{{SD}_{c}}^{2}}}$$

*SNR* was calculated to quantify the signal quality, focusing on ROIc, which represents the critical region adjacent to the esophagus.
$$SNR=\frac{{HU}_{c}}{{SD}_{c}}$$

*BHA* was calculated to assess the impact of beam hardening in ROIc, reflecting the image quality in the region adjacent to the esophagus.
$$BHA=\sqrt{{{SD}_{c}}^{2}-{{SD}_{d}}^{2}}$$

Images acquired with a dilution ratio of 1:24 served as the control group, corresponding to iodine concentrations of 12.8 mgI/mL, 14 mgI/mL, 14.8 mgI/mL, and 16 mgI/mL for the respective contrast agents. The iodine concentration of 12 mgI/mL approximates the concentration of a commercial oral contrast agent (Iohexol, Omnipaque, GE Healthcare, Milwaukee, WI, USA). Objective image quality parameters for the 21 dilution ratios were compared with those of the control group. Image quality similar to the control group was selected: similar image quality was defined as having no statistical differences in at least three of the four objective indexes.

Subjective assessment of image quality: Subjective evaluations were performed by two radiologists, blind to the phantom details, who independently graded the image quality using a 5-point Likert scale (Figure 2). The evaluation criteria were as follows: 1 = extensive artifacts heavily obscure adjacent structural areas, or the difference in density between contrast agents and adjacent structures is indistinguishable; 2 = artifacts are severe, the adjacent structural areas are partially covered, or the difference in density between contrast agents and relatively adjacent structures is difficult to distinguish; 3 = moderate artifacts, the adjacent structure is blurry, or the difference in density between contrast agents and adjacent structures is distinguishable; 4 = minor artifacts, the adjacent structure is well revealed, or the contrast in density between contrast agents and adjacent structures is good; and 5 = artifacts are almost invisible, the adjacent structure is excellently displayed or the density contrast between contrast agents and relatively adjacent structures is excellent.

If there were a discrepancy between the two radiologists, the score would be determined after they had a discussion. A score of four or above was assumed to be of good quality, and a score of three or above was assumed to meet the needs of clinical diagnosis.

### 2.3. Statistical Analysis

Analysis of variance (ANOVA) was used for comparing the objective indexes of image quality among 21 dilution ratios. Multiple comparisons were conducted using the Tamhane test to compare the objective index of image quality between the 21 dilution ratios and the dilution ratio (1:24). The intra-reader reliabilities of objective assessment and subjective assessment were analyzed by using intraclass correlation coefficient (ICC) and Cohen’s Kappa coefficient. The statistical analyses were conducted using SPSS (version 25.0).

## 3. Results

### 3.1. Image Quality Evaluation of Conventional CT Scan

#### 3.1.1. Intra-Reader Agreement and Objective Image Quality Analysis

Intra-reader agreements were excellent for the mean-HU and SD of ROIa, ROIb, ROIc, and ROId (Appendix A). A subjective score of image quality had a good agreement between the two radiologists with κ value of 0.80.

As the dilution ratio increased (leading to a decrease in the iodine concentration of the oral contrast agent), *CNR_a/b_* and *CNR_a/c_* exhibited a biphasic trend, initially increasing due to the reduction of beam-hardening artifacts caused by excessively high concentrations before 1:4. This was followed by a gradual decrease, which became more pronounced beyond 1:24 as the iodine concentration was insufficient to provide adequate signal intensity. *SNR* demonstrated a consistent upward trend, potentially due to the reduction of noise interference with higher dilution ratios, with the rate of change slowing after 1:6. Conversely, the *BHA* around the contrast agent progressively decreased, attributed to reduced iodine attenuation and improved uniformity of the surrounding area, with the rate of change stabilizing after 1:5 (Figure 3). With a specified scanning voltage and iodine concentration specification, the *CNR_a/b_*, *CNR_a/c_*, *SNR*, and *BHA* among the 21 dilution ratios were significantly different. Following multiple comparisons, the dilution ratios that exhibited image quality similar to the control group were summarized in Table 1. These dilution ranges provided an optimal balance across different voltages, ensuring adequate *CNR*, acceptable *SNR*, and minimized *BHA*, as also reflected in the trends observed in Figure 3.

#### 3.1.2. Subjective Image Quality Assessment and Dilution Ratios

After the subjective evaluation of image quality, the dilution ratios of the images that met the requirements for clinical diagnosis (subjective score ≥ 3 points) were summarized in Table 2. Additionally, the dilution ratios that achieved good image quality (subjective score ≥ 4 points) were summarized in Table 3. To achieve higher image quality, it is necessary to narrow the dilution ratio range.

For the control group (a dilution ratio of 1:24), all images met the clinical diagnostic requirements (subjective score ≥ 3 points). The range of dilution ratios that produced objective image quality comparable to the 1:24 ratio (as determined through objective evaluation) was narrower than the range identified through subjective evaluation with scores of three or higher. However, the 1:24 dilution ratio did not consistently achieve a subjective score of four or above. Only when iodine contrast agents of 370 mgI/mL and 400 mgI/mL, diluted at a ratio of 1:24, were used with a CT scan at 80 kVp, and 320 mgI/mL diluted at 1:24 was scanned at 120 kVp, did the subjective score exceed four.

### 3.2. Advantages and Limitations of Dual-Energy CT for CT Esophagography

#### 3.2.1. Additional Value of Dual-Energy CT (DECT) in Improvement of Image Quality

In the dual-energy CT scan images (both kVp-like images and monochromatic energetic images), a significant reduction in *BHA* was observed (Figure 4). The objective image quality was also obviously improved on kVp-like images than kVp images when iodine concentration was high (1:0 to 1:5, Figure 5). At a dilution ratio of 1:1, the image quality of dual-energy CT scans met the basic diagnostic requirements. However, excessively low iodine concentrations after dilution resulted in a slight reduction in image quality on the dual-energy CT scan images. Nonetheless, at a dilution ratio of 1:49, the density contrast in the 100 kVp-like and 70 mono-keV images remained adequate for diagnosis. In summary, dual-energy CT scans accommodate a wider range of dilution ratios (Table 3).

#### 3.2.2. Unsuitability of VUE Images and MAR Reconstruction for CT Esophagography

When the iodine concentration falls to a certain range (dilution ratio ≤ 1:2), the VUE images and reconstruction series with MAR (dilution ratio ≤ 1:1) will undergo contrast subtraction, leading to a reduction in the visualization of the contrast agent. The 100 kVp-like images with MAR enabled at a dilution ratio of 1:1 also demonstrated significant iodine subtraction. Therefore, these imaging protocols are not suitable for CT esophagography examination (Figure 6).

## 4. Discussion

Our literature review revealed significant variability in the reported iodine dilution ratios, ranging from 1:0 to 1:150, as well as inconsistencies in the stock solutions of iodine-based contrast agents. The final iodine concentrations in oral contrast agents span a broad range, from 1.98 mgI/mL to 370 mgI/mL [2,6,7,9,10,11,12,13,14,15,16,17,18,19,20,21] (Appendix A), which undoubtedly impacts the diagnostic accuracy for esophageal perforation. To address this issue, we identified an optimal range of contrast agent concentrations based on experimental data, ensuring compatibility with various scanning conditions, contrast agent concentrations, and dual-energy CT protocols for esophagography. Moreover, this study highlights the advantages of dual-energy CT in enhancing image quality and identifies certain tools that are unsuitable for use in CT esophagography.

### 4.1. Image Quality Evaluation of CT Esophagography

As the dilution ratio increased (corresponding to a decrease in the iodine concentration of the oral contrast agent), *CNR_a/b_* and *CNR_a/c_* exhibited a biphasic trend. An initial increase was observed before the 1:4 dilution ratio, likely due to the reduction of beam-hardening artifacts caused by excessively high iodine concentrations. This finding highlights that moderate reductions in iodine concentration can mitigate beam-hardening effects and improve image clarity. However, beyond the 1:24 dilution ratio, a pronounced decrease in CNR was observed, attributed to insufficient iodine concentration to maintain adequate signal intensity. This underscores the importance of maintaining a minimal iodine threshold to ensure diagnostic image quality.

*SNR* demonstrated a consistent upward trend with increasing dilution ratios, potentially due to reduced noise interference. However, the rate of increase plateaued beyond 1:6, suggesting diminishing benefits at higher dilution ratios. Conversely, the *BHA* around the contrast agent progressively decreased, with its rate of change stabilizing after 1:5. These findings indicate that lower iodine concentrations can effectively reduce artifacts while maintaining image uniformity. The maximum *BHA* value (52.31) and minimum *SNR* value (0.32) identified in this study were associated with high-quality images (subjective score ≥ 4 points) and may serve as benchmarks for assessing image quality in CT esophagography (Figure 3).

Across the 21 dilution ratios, the metrics of *CNR_a/b_*, *CNR_a/c_*, *SNR*, and *BHA* were significantly different under various scanning voltages and iodine concentrations. Following multiple comparisons, the dilution ratios that provided image quality comparable to the control group were summarized in Table 1. These ranges represent an optimal balance between *CNR*, *SNR*, and minimized *BHA* across different voltages, as reflected in the trends observed in Figure 3.

### 4.2. Dilution Recommendations for the Preparation of Oral Contrast Agent in CT Esophagography

In this study, we aimed to identify the optimal dilution ratios for different iodine contrast agent concentrations and scanning conditions, using both objective and subjective image quality evaluations. The suggested dilution ratios for achieving high-quality images in CT esophagography should be adjusted based on the iodine concentration of the contrast agent stock solution and the scanning voltages used. Table 1, Table 2 and Table 3 demonstrate how suitable dilution ranges vary depending on these factors, with objective evaluations summarized in Table 1 and subjective assessments in Table 2 and Table 3.

By using a dilution ratio of 1:24 as the control group for iodine concentration approximated to a commercial oral contrast (12 mgI/mL), the appropriate dilution ratios producing comparable image quality parameters were identified through objective image quality evaluation. The range of ratios identified through objective evaluation was narrower compared to those identified through subjective evaluation, which met the basic clinical diagnostic requirements (subjective score ≥ 3). Furthermore, the 1:24 dilution ratio did not always achieve a subjective score of four or above, suggesting that the range of ratios obtained from objective evaluation may not reliably predict high-quality images. To contextualize our findings, we compared the dilution ratios identified in our study with those reported in the literature, highlighting both consistencies and differences. From the perspective of dilution ratios, half of the studies (8 out of 16) [7,9,10,11,13,14,18,20] (Appendix A) aligned with the dilution ratios we recommend for meeting basic diagnostic needs, corresponding to image quality scores of three or higher. However, our study emphasizes that the preparation of contrast agents should not only focus on iodine concentration but also account for variations in scanning conditions. These findings provide meaningful guidance for standardizing image quality in CT esophagography across diverse clinical practices.

To ensure excellent image quality, dilution ratios that achieved superior image quality, corresponding to a subjective score of ≥4 were identified in the objective evaluation. Based on these dilution ratios (subjective score ≥ 4), specific recommendations for preparing 100 mL of oral contrast agent solution are listed in Table 4. These suggestions can help ensure optimal image quality during CT examinations.

Additionally, this highlights the importance of incorporating both image quality evaluation methods to ensure diagnostic confidence in clinical settings. Future studies need to explore how to enhance the correlation between these two methods, ensuring that objective parameters can more closely predict subjective clinical preferences.

### 4.3. Additional Value of DECT in Improving Image Quality

DECT-derived kVp-like images and monochromatic images effectively reduced *BHAs*, enhancing image quality and accommodating a broader range of dilution ratios.

The reduction of *BHA* on DECT-derived monochromatic images has been reported, and selecting appropriate monochromatic energy levels can further enhance this effect [25]. The present study found that monochromatic images at higher keV levels demonstrated a more pronounced reduction in *BHAs* (Figure 4). However, low iodine concentrations after dilution resulted in a decrease in density contrast. In both conventional scans and dual-energy scans, compared to series at higher energy levels (120 kVp, 120 kVp-like, and 90 mono-keV), series at lower levels (100 kVp, 100 kVp-like and 70 mono-keV) exhibited more pronounced density contrast and can meet the diagnostic requirement with a dilution rate of 1:49. However, image noise increased on monochromatic images with lower keV levels, such as 40 keV and 60 keV. Compared to conventional scans, dual-energy scans are not helpful in enhancing the density contrast when the iodine concentration is too low.

Dual-energy scans with MAR undergo contrast subtraction at a specific concentration, leading to a reduction in the visualization of the contrast agent; this may be the reason that the contrast agent at a specific iodine concentration is similar to metal in CT attenuation. VNE can identify the contrast agent at a specific iodine concentration range and replace it with a non-iodized density, which is also not suitable for CT esophagography examination.

Our study has a few limitations that need to be considered. Firstly, the phantom experiment may not fully replicate the complex anatomical environment and inflammatory states of the human body. Additionally, the study has not yet been applied to patients; therefore, its findings remain preliminary. Future work will involve animal experiments to validate these findings and prospective clinical studies to evaluate the diagnostic efficacy of CT esophagography in detecting esophageal perforations and its utility in subsequent follow-up.

## 5. Conclusions

This study reliably identified the optimal dilution ratios for oral contrast agents in CT esophagography through both objective and subjective evaluations, considering variations in contrast media types and scanning conditions across institutions. The findings provide valuable reference ranges for both conventional and dual-energy CT scanning, addressing clinical needs in diverse practice settings. These results contribute to improving the practice of CT esophagography and support the broader implementation of dual-energy CT in clinical applications.

## Figures and Tables

**Figure 1 bioengineering-11-01300-f001:**
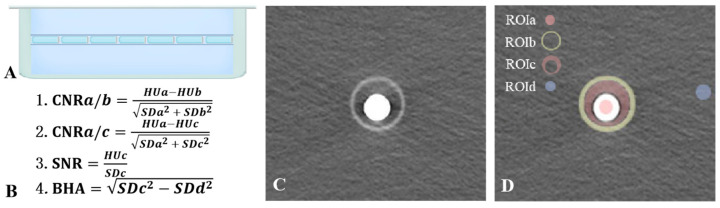
Schematic diagram of phantom and related formula. (**A**) Schematic diagram of phantom. (**B**) Computational formulas of objective indexes. (**C**) Primal CT image obtained through experiments. (**D**) Schematic diagram of ROI positions. ROIa, ROIb, ROIc, and ROId, respectively, represent the area of the contrast agent in Eppendorf tubes, the wall of hollow plastic tubes, the area between the contrast agent in Eppendorf tube and the peripheral plastic tube, and the background.

**Figure 2 bioengineering-11-01300-f002:**
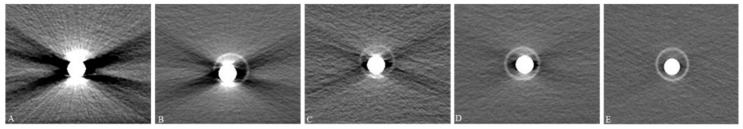
Example of subjective scoring of CT images. (**A**) Extensive artifacts heavily obscure adjacent structural areas, graded 1 point. (**B**) severe artifacts partially cover the adjacent structural areas, graded 2 points. (**C**) moderate artifacts, adjacent structure is blurry, graded 3 points. (**D**) minor artifacts, the adjacent structure is well revealed, graded 4 points. (**E**) artifacts are almost invisible, adjacent structure is excellent displayed, graded 5 points.

**Figure 3 bioengineering-11-01300-f003:**
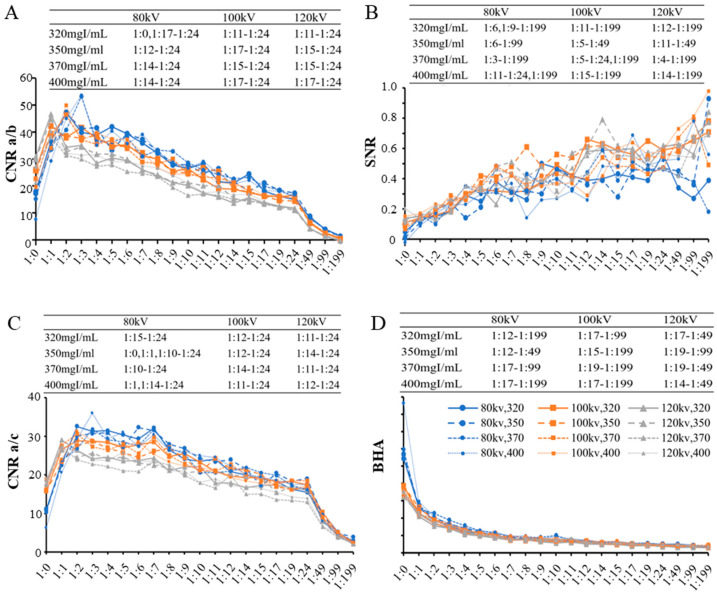
Line charts show trends of the objective index of image quality. (**A**) *CNR_a/b_*; (**B**) *SNR*; (**C**) *CNR_a/c_*; (**D**) *BHA*. Note: after multiple comparisons, the data table (upper right) summarizes the range of contrast agent dilution ratios which can achieve similar image quality to the control group. *CNR*: contrast-to-noise ratio; *SNR*: signal-to-noise ratio; *BHA*: beam-hardening artifact.

**Figure 4 bioengineering-11-01300-f004:**
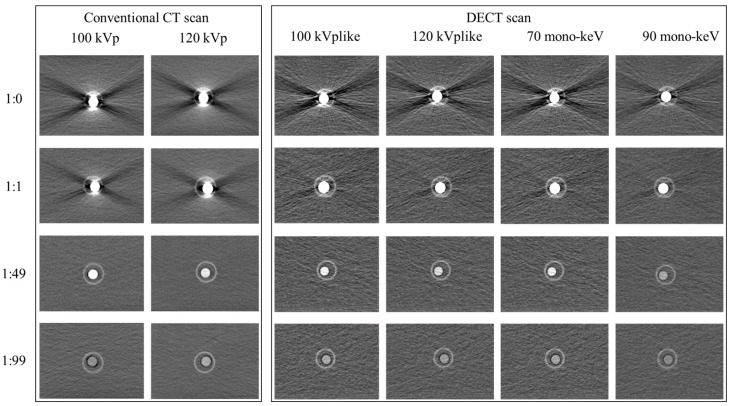
Comparison of image quality among images obtained by conventional scan and dual-energy scan. DECT: dual-energy CT.

**Figure 5 bioengineering-11-01300-f005:**
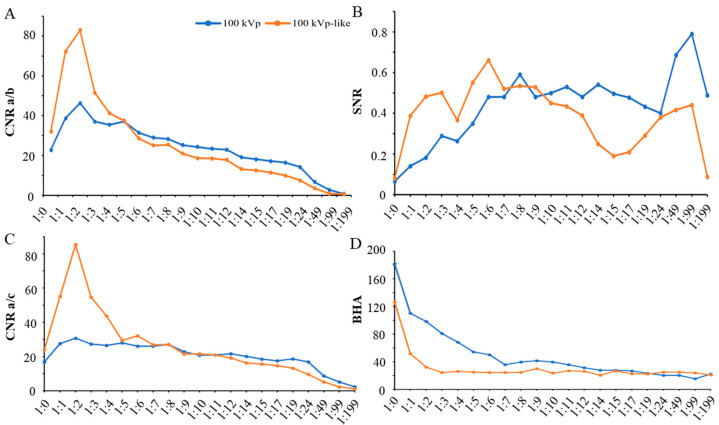
Line graphs of the objective index of image quality underwent conventional CT scan at 100 kVp and dual-energy scan with 100 kVp-like reconstruction. (**A**) *CNR_a/b_*; (**B**) *SNR*; (**C**) *CNR_a/c_*; (**D**) *BHA*. *CNR*: contrast-to-noise ratio; *SNR*: signal-to-noise ratio; *BHA*: beam-hardening artifact.

**Figure 6 bioengineering-11-01300-f006:**
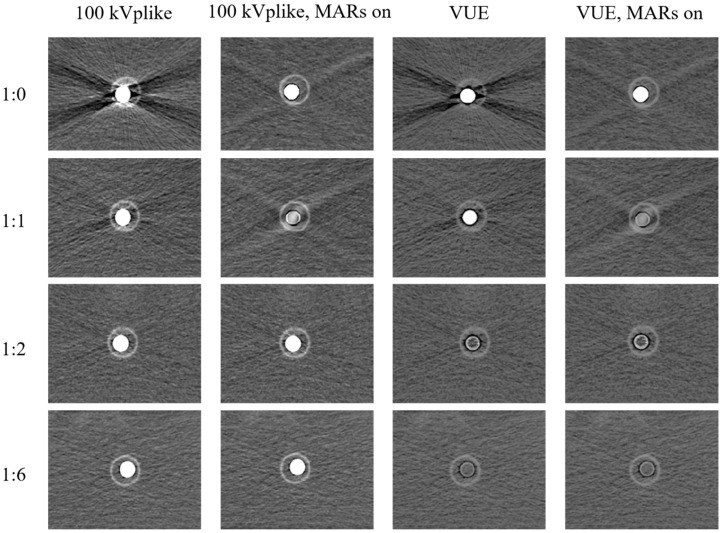
Comparation of image quality among images obtained by dual-energy scan. VNE: virtual non-enhanced; MAR: metal artifact reduction.

**Table 1 bioengineering-11-01300-t001:** Dilution ratios that had image quality similar to the control group (1:24).

Concentration Specifications	80 kVp	100 kVp	120 kVp
320 mgI/mL	1:15~1:24	1:12~1:24	1:12~1:24
350 mgI/mL	1:12~1:24	1:15~1:24	1:15~1:24
370 mgI/mL	1:14~1:24	1:15~1:24	1:15~1:24
400 mgI/mL	1:14~1:24	1:17~1:24	1:14~1:24

**Table 2 bioengineering-11-01300-t002:** Dilution that met the requirements for clinical diagnosis (subjective score ≥ 3 points).

Concentration Specifications	80 kVp	100 kVp	120 kVp
320 mgI/mL	1:4~1:49	1:4~1:49	1:3~1:24
350 mgI/mL	1:6~1:49	1:4~1:49	1:3~1:24
370 mgI/mL	1:6~1:49	1:4~1:49	1:3~1:24
400 mgI/mL	1:6~1:49	1:4~1:49	1:3~1:24

**Table 3 bioengineering-11-01300-t003:** Dilution ratios that achieved good image quality (subjective score ≥ 4 points).

Concentration Specifications	80 kVp	100 kVp	120 kVp	Dual-Energy Scan 80/140 (Fast Switching)
320 mgI/mL	1:6~1:19	1:6~1:19	1:5~1:24	1:1~1:24
350 mgI/mL	1:10~1:19	1:8~1:19	1:6~1:19	1:1~1:24
370 mgI/mL	1:10~1:24	1:8~1:19	1:6~1:19	1:1~1:24
400 mgI/mL	1:11~1:24	1:8~1:19	1:6~1:19	1:1~1:24

**Table 4 bioengineering-11-01300-t004:** Specific recommendations for preparing 100 mL of the oral contrast solution.

Scanning Voltage	Concentration Specifications ^1^	Volume ^2^
80 kV	320 mgI/mL 320 mgI/mL, 370 mgI/mL 400 mgI/mL	5~14 mL 5~9 mL 5~8 mL
100 kV	320 mgI/mL 350 mgI/mL, 370 mgI/mL, 400 mgI/mL	5~14 mL 5~11 mL
120 kV	320 mgI/mL 350 mgI/mL, 370 mgI/mL, 400 mgI/mL	5~16 mL 5~14 mL

^1^ Concentration specifications of the iodine contrast agent stock solution. ^2^ The volume of contrast agent stock solution. Note: 1. The volume of iodine contrast agent stock solution should not be less than 5 mL. 2. While the maximum allowable volume of iodine contrast agent stock solution should consider the scanning voltage and the concentration specifications, taking a tube voltage of 100 kVp as an example, for a concentration of 320 mgI/mL, the volume of contrast agent stock solution should not exceed 14 mL; for concentrations of 350 mgI/mL, 370 mgI/mL, and 400 mgI/mL, the volume of contrast agent stock solution should not exceed 11 mL.

## Data Availability

The datasets generated for this study can be found in the submitted article.

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
