# Peer review of "Optimizing CT Esophagography: Ex Vivo Study on Contrast Ratios, Image Quality, and Dual-Energy Benefits"

_bioengineering, 2024, doi:10.3390/bioengineering11121300_

Round 1

Reviewer 1 Report

Comments and Suggestions for Authors

This paper aims to optimize CT esophagography using ex-vivo study on contrast ratios, image quality, and dual-energy benefits.

I have some comments:

1. The equations in Figure should be defined and explained in text.

2. I don't see any optimal results.

3. The results should be compared with existing paper published in recent years.

4. If there is no optimal results or methods, the title should be changed.

Comments on the Quality of English Language

Quality is ok.

Author Response

Reviewer 1

This paper aims to optimize CT esophagography using ex-vivo study on contrast ratios, image quality, and dual-energy benefits.

I have some comments:

Comments 1: The equations in Figure should be defined and explained in text.

Response 1:

Thank you for your valuable suggestion. We have now updated the manuscript to include detailed explanations of the equations in the figure within the text.

“Objective evaluations were conducted on the acquired images by measuring parameters related to CT attenuation and image noise. Three axial slices without air bubbles were selected from each Eppendorf tube by an observer. Four distinct regions of interest (ROIs) were segmented (Figure 1) independently by two radiologists using ImageJ software (National Institutes of Health, Bethesda, MD, USA),

ROIa: The contrast agent within the Eppendorf tube, representing esophageal contrast material with high CT attenuation.

ROIb: The wall of the hollow plastic tube, mimicking other slightly high-density tissues in the mediastinum.

ROIc: The transition zone between the contrast agent and the surrounding plastic tube, simulating the region adjacent to the esophagus in a clinical setting.

ROId: The background region outside the tube, representing a reference background area required for image quality analysis.

CT attenuation (Hounsfield Unit, HU) and SD (image noise, HU) of ROIs were measured. Objective parameters of image quality included[23,26]: Contrast-to-Noise Ratio (CNR), Signal-to-Noise Ratio (SNR), and Beam-Hardening Artifact (BHA). These parameters, which are commonly used for objective image quality assessment, were calculated using the following formulas:

CNR was calculated to evaluate the contrast between specific regions of interest: CNRa/b: Represents the contrast between esophageal contrast material (ROIa) and mediastinal fluid (ROIb).

... ...

(showed in Manuscript P3-4, L99-130)

Each equation is now clearly defined and its relevance to the image quality parameters has been explained to enhance clarity for the readers. We appreciate your suggestion and believe this revision improves the overall comprehensibility of the manuscript.

Comments 2: I don't see any optimal results.

Response 2:

Thank you for your valuable question. The use of oral contrast agents in CT esophagography protocols varies widely, spanning from undiluted to 200-fold dilution, as reported in the literature. Our results provide optimized recommendations for the use of oral contrast agents in CT esophagography. These recommendations consider variations in scanning conditions and the initial concentrations of contrast agents, offering meaningful guidance. Additionally, for dual-energy CT, we not only suggest appropriate dilution ratios but also identify tools that are unsuitable for this application, which supports the broader adoption of dual-energy CT in clinical practice.

During the revision process, we realized that your concern may have arisen due to insufficient clarity in several parts of the manuscript. To address this, we have made extensive revisions, including:

Methods: We clarified that the reference dilution ratios for contrast agents were derived from the literature(P2, L79-81). Additionally, we elaborated on the parameters used in the equations for phantom experiments, explaining their significance(P3-4, L99-130). 

Results: We provided further explanation of the objective parameter evaluations, emphasizing that they represent more than a simple trend(P4-5, L167-176,L180-182).

Discussion: We expanded the comparison between our experimental results and findings from the literature, highlighting the significance of our study(P9, L292-300) . We summarized the key results to emphasize their importance in the first paragraph(P8, L238-247) and further analyzed the impact of different dilution ratios of oral contrast agents on image quality in CT esophagography (P8-9, L249-275) .

Conclusion: We refined the wording to provide a more precise and focused summary of the main findings.(P10, L348-354)

Comments 3:The results should be compared with existing papers published in recent years.

Response 3:

Thank you for your suggestion. Based on your recommendation, we have supplemented the Methods section with a summary table (Table S1) detailing the oral contrast agent protocols reported in recent literature, which also served as a basis for determining the dilution ratios used in our experiments. Additionally, we expanded the Discussion section to include comparisons between our findings and those reported in previous studies.

Specifically, we have expanded the Discussion section to include a comparative analysis of oral contrast agents reported in the literature.

"To contextualize our findings, we compared the dilution ratios identified in our study with those reported in the literature, highlighting both consistencies and differences. From the perspective of dilution ratios, half of the studies (8 out of 16) (Table S1)aligned with the dilution ratios we recommend for meeting basic diagnostic needs, corresponding to image quality scores of 3 or higher. However, our study emphasizes that the preparation of contrast agents should not only focus on iodine concentration but also account for variations in scanning conditions. These findings provide meaningful guidance for standardizing image quality in CT esophagography across diverse clinical practices."(P9, L292-300)

Comments 4: If there is no optimal results or methods, the title should be changed.

Response 4:

Thank you for your feedback. We believe the title does not require modification. The use of oral contrast agents in CT esophagography protocols varies widely, spanning from undiluted to 200-fold dilution, as reported in the literature. In this manuscript, our results indeed provide optimized recommendations for the use of oral contrast agents in CT esophagography. These recommendations consider variations in scanning conditions and the initial concentrations of contrast agents and provide guidance based on meeting image quality and diagnostic needs. Additionally, for dual-energy CT, we not only suggest appropriate dilution ratios but also identify tools that are unsuitable for this application, which supports the broader adoption of dual-energy CT in clinical practice.

We understand that the perception of a lack of optimal results may have arisen from insufficient emphasis in the results, discussion, and conclusion sections. Therefore, we have revised these sections, consistent with the revisions made in response to the second question, to better align with and highlight the significance of our findings as reflected in the title.

Reviewer 2 Report

Comments and Suggestions for Authors

Review for the manuscript:

Entitled: "Optimizing CT Esophagography: Ex-Vivo Study on Contrast Ratios, Image Quality, and Dual-Energy Benefits"

for Bioengineering.

With ID: bioengineering-3271980

General comments

Comments for the Authors

This work is well within the scope of Bioengineering, and it may be of interest to particular readers of this journal, with good references to follow. The topic is interesting and novel since the use of dual-energy CT has not been yet explored in order to improve image quality for CT esophagography. However, the structure of the manuscript need some more effort, since various paragraphs are repeated and the preliminary findings of this work are not disseminated in a clear and straightforward manner.

 For all the above, and the specific comments below, I have opted to recommend a Major revision for the current form of this work.

Specific comments

Introduction

P1, L40: ‘of life. evaluation is crucial.’ Please correct.

Materials and Methods

P3, Fig.1: ‘D: Schematic diagram of ROIs position’. The ‘D’ letter does not appear in the figure. Please fix it.

P3, L108: ‘ImageJ’ Please add company and country information, following the instruction to authors.

P3, L109-112: ‘Objective parameters of image quality included:’ This was stated a couple of times earlier, like in P3,L90-93. As well as ‘Four iodine contrast agents (320 mgI/mL, 350 mgI/mL, 370 mgI/mL, 400 mgI/mL)’ in P3, L113. Please revise.

P6, L202: ‘thickness of 2mm’ Please explain the selection of this thickness.

P6, L206: ‘detectors is CsI:Tl (Thallium doped cesium iodide) crystal’ Please specify the thickness of the scintillator.

Results

P4, L152: ‘As the dilution ratio changed (the iodine concentration of oral contrast decreased), CNRa/b and CNRa/c initially increased and then decreased. SNR demonstrated an overall upward trend. The BHA around the contrast agent showed a downward trend’ Please revise this section. Decreased, increased, decreased, does not explain scientifically the CNR results. 

Figures 3 and 5 are of low quality and are not clearly visible. Please revise.

P4, L156: ‘Following multiple comparisons, the dilution ratios that exhibited image quality similar to the control group were summarized in table1.’ The discussion of Table 1 should be updated with more details.

Fig.4 and Fig.6. Please revise the small ‘v’ in every instance of ‘kvp’-> ‘kVp’ and ‘kev’-> ‘keV’.

P7, L207: ‘on at dilution ratio of 1:1 also showed iodine subtraction’ Please revise.

Discussion

P8, L221: ‘CNRa/b and CNRa/c initially increased and then decreased. SNR showed an overall upward trend while BHA had a downward trend with increasing dilution ratio.’ Please elaborate more on this.

P8, L239: ‘tion. and the range’ Please correct.

Conclusions

Please provide some more effort to highlight the findings and the importance of this work.

Author Response

Reviewer 2

General comments

Comments for the Authors

This work is well within the scope of Bioengineering, and it may be of interest to particular readers of this journal, with good references to follow. The topic is interesting and novel since the use of dual-energy CT has not been yet explored in order to improve image quality for CT esophagography. However, the structure of the manuscript need some more effort, since various paragraphs are repeated and the preliminary findings of this work are not disseminated in a clear and straightforward manner.

For all the above, and the specific comments below, I have opted to recommend a Major revision for the current form of this work.

Response:

Thank you very much for reviewing our manuscript and providing constructive comments. We have carefully read them and made specific adjustments to address your concerns. Please find the details below.

Specific comments

Comments 1: Introduction

P1, L40: ‘of life. evaluation is crucial.’ Please correct.

Response 1: Thank you for pointing out the error. We have corrected it:

"Early radiological diagnosis and accurate post-treatment assessment are crucial for improving patient outcomes, reducing hospital stay duration and costs, and enhancing quality of life."(P1, L40) 

This revision removes redundancy and improves readability.

Comments 2: Materials and Methods

P3, Fig.1: ‘D: Schematic diagram of ROIs position’. The ‘D’ letter does not appear in the figure. Please fix it.

Response 2: Thank you for pointing out this issue. During the manuscript revision process, we updated Figure 1 to better illustrate the ROI delineation by including the original image. However, the updated figure was inadvertently not included in the final submission, resulting in a mismatch between the figure and its legend. We have now replaced the figure with the correct version(P2, L86) , ensuring that all elements, including ‘D,’ are consistent with the figure legend. We appreciate your careful review and apologize for this oversight.

Comments 3: P3, L108: ‘ImageJ’ Please add company and country information, following the instruction to authors.

Response 3: Thank you for your comment. We have updated the text to include the company and country information as follows: "ImageJ software (National Institutes of Health, Bethesda, MD, USA)."(P3, L103)  We appreciate your attention to this detail.

Comments 4: P3, L109-112: ‘Objective parameters of image quality included:’ This was stated a couple of times earlier, like in P3,L90-93. As well as ‘Four iodine contrast agents (320 mgI/mL, 350 mgI/mL, 370 mgI/mL, 400 mgI/mL)’ in P3, L113. Please revise.

Response 4: Thank you for pointing out this omission. We have now revised the methodology section(P3, L99-130) by removing redundant information and adding the necessary equations and related explanations. This revision clarifies the objective parameters of image quality and eliminates repetition. We appreciate your careful review and suggestions.

Comments 5: P6, L202: ‘thickness of 2mm’ Please explain the selection of this thickness.

Response 5: Thank you for pointing this out. Upon reviewing the manuscript, we realized that the reference to "2mm slice thickness" was unclear. In fact, all CT images used in our study were acquired with a slice thickness of 1.25mm. This information has been provided in the supplementary material. Thank you for your careful review.

Comments 6: P6, L206: ‘detectors is CsI:Tl (Thallium doped cesium iodide) crystal’ Please specify the thickness of the scintillator.

Response 6: Thank you for your comment. However, we did not mention the use of CsI:Tl (Thallium doped cesium iodide) crystals or specify their thickness in our manuscript. It seems there might have been a misunderstanding. Could you please confirm if this is related to another section of the manuscript or a different reference? We would be happy to clarify or make the necessary adjustments if needed.

Comments 7: Results

P4, L152: ‘As the dilution ratio changed (the iodine concentration of oral contrast decreased), CNRa/b and CNRa/c initially increased and then decreased. SNR demonstrated an overall upward trend. The BHA around the contrast agent showed a downward trend’ Please revise this section. Decreased, increased, decreased, does not explain scientifically the CNR results. 

Response 7: Thank you for pointing out the need to clarify the scientific explanation of the CNR results. We have revised the section to provide a more detailed interpretation of the trends observed in CNRa/b and CNRa/c as follows:

Revised Text:

"As the dilution ratio increased (leading to a decrease in the iodine concentration of the oral contrast agent), CNRa/b and CNRa/c exhibited a biphasic trend, initially increasing due to the reduction of beam-hardening artifacts caused by excessively high concentrations before 1:4. This was followed by a gradual decrease, which became more pronounced beyond 1:24 as the iodine concentration was insufficient to provide adequate signal intensity. SNR demonstrated a consistent upward trend, potentially due to the reduction of noise interference with higher dilution ratios, with the rate of change slowing after 1:6. Conversely, the BHA around the contrast agent progressively decreased, attributed to reduced iodine attenuation and improved uniformity of the surrounding image, with the rate of change stabilizing after 1:5."(P4-5, L167-176)

This revision addresses the scientific reasoning behind the observed changes in CNR and provides a more comprehensive explanation of the trends. We hope this adequately clarifies the results.

Comments 8: Figures 3 and 5 are of low quality and are not clearly visible. Please revise.

Response 8: Thank you for pointing out the quality issue with Figures 3 and 5. We sincerely apologize for the oversight. We have revised and replaced both figures with higher-resolution versions to ensure clarity and visibility of the data. These updated figures have been carefully checked to meet publication standards and to enhance readability. Please let us know if further adjustments are required.

Comments 9: P4, L156: ‘Following multiple comparisons, the dilution ratios that exhibited image quality similar to the control group were summarized in table1.’ The discussion of Table 1 should be updated with more details.

Response 9: Thank you for your valuable feedback. In response, we have added emphasis in the results section, highlighting that These dilution ranges provided an optimal balance across different voltages, ensuring adequate contrast, acceptable signal-to-noise ratios, and minimized beam-hardening artifacts, as also reflected in the trends observed in Figure 3.(P5, L180-182) Additionally, we have revised the results section related to Figure 3 (P5, L167-176) to emphasize specific key dilution ratios, such as the significant decline in CNR observed beyond the 1:24 ratio. This trend aligns with the recommended ranges in Table 1, further demonstrating that this range ensures optimal image quality.

Moreover, the subsequent sections of the results compared subjective and objective evaluation outcomes. Notably, while the final recommended ranges were derived from subjective evaluation results, our study also emphasizes in the discussion section the importance of integrating both subjective and objective methods for image quality assessment. This combined approach ensures greater confidence in clinical diagnostics. Future studies should aim to enhance the correlation between these two methods to ensure that objective parameters can more accurately predict subjective clinical preferences.

Comments 10: Fig.4 and Fig.6. Please revise the small ‘v’ in every instance of ‘kvp’-> ‘kVp’ and ‘kev’-> ‘keV’.

Response 10: Thank you for pointing out the inconsistencies in the capitalization of ‘kVp’ and ‘keV’ in Figures 4 and 6. We have carefully reviewed and revised all instances to ensure the correct usage of ‘kVp’ and ‘keV’ throughout the figures. The updated versions have been uploaded for your review. Please let us know if there are any additional concerns.

Comments 11: P7, L207: ‘on at dilution ratio of 1:1 also showed iodine subtraction’ Please revise.

Response 11: Thank you for pointing out the need for revision. We have clarified and revised the sentence for better readability and accuracy. The updated text now reads:

"The 100 kVp-like images with MARs enabled at a dilution ratio of 1:1 also demonstrated significant iodine subtraction."(P7, L230-232) 

This revision ensures clarity and maintains the scientific accuracy of the statement. Please let us know if there are any further suggestions.

Comments 12: Discussion

P8, L221: ‘CNRa/b and CNRa/c initially increased and then decreased. SNR showed an overall upward trend while BHA had a downward trend with increasing dilution ratio.’ Please elaborate more on this.

Response 12: Thank you for your insightful comment. We have revised and elaborated to provide a more detailed explanation of the observed trends in the section 4.1 Image quality evaluation of CT esophagography in the discussion.

Revised Text:

As the dilution ratio increased (leading to a decrease in the iodine concentration of the oral contrast agent), CNRa/b and CNRa/c exhibited a biphasic trend. The initial increase, observed before the 1:4 dilution ratio, was attributed to the reduction of beam-hardening artifacts caused by excessively high iodine concentrations. This suggests that lower iodine concentrations can mitigate the negative effects of beam-hardening, enhancing image clarity. Beyond the 1:24 dilution ratio, a pronounced decrease was observed, likely due to insufficient iodine concentration to provide adequate signal intensity, which highlights the importance of maintaining a minimal threshold of iodine concentration for diagnostic purposes.

SNR demonstrated a consistent upward trend, potentially due to reduced noise interference with higher dilution ratios, and the rate of change plateaued after 1:6. Conversely, the BHA around the contrast agent progressively decreased, with the rate of change stabilizing after 1:5. These trends indicate that lower iodine concentrations can effectively minimize artifacts without compromising uniformity. The maximum BHA value (52.31) and minimum SNR value (0.32) identified in this study, associated with high-quality images (subjective score ≥4 points), may serve as benchmarks for assessing image quality in CT esophagography (Figure 3).

With a specified scanning voltage and iodine concentration specification, the metrics of CNRa/b, CNRa/c, SNR, and BHA across the 21 dilution ratios were significantly different. Following multiple comparisons, the dilution ratios that provided image quality comparable to the control group were summarized in Table 1. These ranges represent an optimal balance across different voltages, ensuring adequate contrast, acceptable signal-to-noise ratios, and minimized beam-hardening artifacts, as reflected in Figure 3.(P8-9, L249-275)

We believe this revised section provides a more comprehensive and clinically relevant explanation of the results and adequately addresses the reviewer's concerns. 

Comments 13: P8, L239: ‘tion. and the range’ Please correct.

Response 13: Thank you for pointing out this issue. We have corrected the inconsistency in capitalization. The phrase "evaluation. and the range" has been revised to "evaluation. The range" (P9, L288) to ensure proper formatting and alignment with standard writing conventions.

Comments 14: Conclusions

Please provide some more effort to highlight the findings and the importance of this work.

Response 14: Thank you for your suggestion to enhance the conclusions and better emphasize the findings and importance of this work. We have revised the conclusion section to provide a clearer and more impactful summary of the study's contributions, as follows:

"This study reliably identified the optimal dilution ratios for oral contrast agents in CT esophagography through both objective and subjective evaluations, considering variations in contrast media types and scanning conditions across institutions. The findings provide valuable reference ranges for both conventional and dual-energy CT scanning, addressing clinical needs in diverse practice settings. These results contribute to improving the practice of CT esophagography and support the broader implementation of dual-energy CT in clinical applications."(P10, L348-354)

This revised conclusion highlights the key findings of the study, underscores their relevance in clinical practice, and reflects the novelty and importance of the work in advancing CT esophagography techniques. We appreciate your constructive feedback and believe this revision addresses your concern effectively.

Reviewer 3 Report

Comments and Suggestions for Authors

Dear Editor and Authors,

Thank you for asking me to review this experimental work titled “Optimizing CT Esophagography: Ex-Vivo Study on Contrast Ratios, Image Quality, and Dual-Energy Benefits” by Dr. Hao and colleagues from Wuhan, China.

This is a straight forward and although one may say simplistic study, quite significant in developing an optimal scanning strategy.  The authors have correctly identified the need of investigating the optimal dosage/dilution concentration of oral iodine contrast medium for performing CT esophagography to better identify perforations. Currently, there is variability amongst institutions!

They utilized a mock esophagus constructed from a two-layered tube-like phantom and utilized various dosage/dilution score. They obtained and compared both objective and subjective measurements.

The methodology and conduct of the study is quite good with a robust modeling approach, and thorough measurements. They determined that the optimal dilution ratios are 1:6 to 1:19 for 320 mgI/mL, and 1:8 to 1:19 for 350 to 400 mgI/mL.

I have some minor comments to make:

1.       Why and how where the dosages/dilutions selected? Is there literature to support this?

2.       Has the modeling setup utilized been validated / approved? Is it robus and clinically applicable?

Overall this is an interesting work that needs some minor editing prior to publication. Thank you and kind regrds!

Comments on the Quality of English Language

Needs some minor language editing!

Author Response

Reviewer 3

Dear Editor and Authors,

Thank you for asking me to review this experimental work titled “Optimizing CT Esophagography: Ex-Vivo Study on Contrast Ratios, Image Quality, and Dual-Energy Benefits” by Dr. Hao and colleagues from Wuhan, China.

This is a straight forward and although one may say simplistic study, quite significant in developing an optimal scanning strategy. The authors have correctly identified the need of investigating the optimal dosage/dilution concentration of oral iodine contrast medium for performing CT esophagography to better identify perforations. Currently, there is variability amongst institutions!

They utilized a mock esophagus constructed from a two-layered tube-like phantom and utilized various dosage/dilution score. They obtained and compared both objective and subjective measurements.

The methodology and conduct of the study is quite good with a robust modeling approach, and thorough measurements. They determined that the optimal dilution ratios are 1:6 to 1:19 for 320 mgI/mL, and 1:8 to 1:19 for 350 to 400 mgI/mL.

Response:

Thank you very much for your constructive comments and affirmation, as well as your encouragement regarding our manuscript. 

 I have some minor comments to make:

Comments 1: Why and how where the dosages/dilutions selected? Is there literature to support this?

Response 1: Thank you for pointing out this important aspect, which helps enhance the clarity and rigor of our methodology. As stated in the Materials and Methods section, the dilution ratios were selected based on clinical practice and relevant literature. To provide further clarity, we have included a new table summarizing the details of oral contrast agents and their dilution ratios reported in previous studies in Table S1. This table aims to highlight the existing variability in dilution concentrations and justify the range of dilutions tested in our study.

Comments 2: Has the modeling setup utilized been validated / approved? Is it robus and clinically applicable?

Response 2: Thank you for raising this important issue. Image quality studies are typically conducted by directly placing multiple contrast agents into commercial water phantoms to assess image quality parameters. In this study, to simulate the esophagus and surrounding environment, and to avoid the mutual influence of contrast agents placed horizontally at the same time, a simplified phantom device was specifically designed. The basic principle behind the design of this phantom is consistent with the water model, making it a reasonable and appropriate choice for our research objectives.

To help readers better understand, we have expanded the description of the model in the methods section and further clarified the significance of each image quality parameter and how it is evaluated(P3, L99-130). These additions are intended to provide readers with more background information, fully demonstrating the scientific validity of the model and its application value in the research.

Round 2

Reviewer 1 Report

Comments and Suggestions for Authors

Accept in present form.

Reviewer 2 Report

Comments and Suggestions for Authors

Review for the manuscript:

Entitled: "Optimizing CT Esophagography: Ex-Vivo Study on Contrast Ratios, Image Quality, and Dual-Energy Benefits"

for Bioengineering.

With ID: bioengineering-3271980.R1

General comments

My previous comments were addressed; thus, the manuscript can be published.

Best regards